# Oral rabies vaccination of dogs—Experiences from a field trial in Namibia

**Conrad Martin Freuling** [1☯]*, **Frank Busch**[2☯], **Adriaan Vos**[3], **Steffen Ortmann**[3], **Frederic Lohr**[4], **Nehemia Hedimbi**[5], **Josephat Peter**[6], **Herman Adimba Nelson**[7], **Kenneth Shoombe**[8], **Albertina Shilongo**[9], **Brighton Gorejena**[10], **Lukas Kaholongo**[10], **Siegfried Khaiseb**[11], **Jolandie van der Westhuizen**[11], **Klaas Dietze**[2], **Goi Geurtse**[12], **Thomas Müller**[1]

1 Institute of Molecular Virology and Cell Biology, Friedrich-Loeffler-Institut (FLI), WHO Collaborating Centre for Rabies Surveillance and Research, WOAH Reference Laboratory for Rabies, Greifswald-Insel Riems, Germany, 2 Institute of International Animal Health/One Health, Friedrich-Loeffler-Institut (FLI), Greifswald-Insel Riems, Germany, 3 Ceva Innovation Center GmbH, Dessau–Rosslau, Germany, 4 Mission Rabies, Cranborne, United Kingdom, 5 Animal Disease Control, Kunene, Ministry of Agriculture, Water & Land Reform, Directorate of Veterinary Services, State Veterinary Office, Opuwo, Namibia, 6 Directorate of Veterinary Services, State Veterinary Office, Omusati, Namibia, 7 Directorate of Veterinary Services, State Veterinary Office, Ondangwa, Namibia, 8 Deputy Chief Veterinary Officer, Animal Disease Control, North, Ministry of Agriculture, Water & Land Reform, Directorate of Veterinary Services, State Veterinary Office, Ongwediva, Namibia, 9 Chief Veterinary Officer, Directorate of Veterinary Services, Ministry of Agriculture, Water and Land Reform, Windhoek, Namibia, 10 Faculty of Agriculture and Natural Resources, Ogongo Campus, University of Namibia, Wnindhoek, Namibia, 11 Central Veterinary Laboratory, Directorate of Veterinary Services (DVS), Ministry of Agriculture Water and Land Reform, Windhoek, Namibia, 12 SWAVET, Windhoek, Namibia

☯ These authors contributed equally to this work.

* conrad.freuling@fli.de

**Data Availability Statement:** All relevant data are within the manuscript and its Supporting Information files.

## Abstract

Dog-mediated rabies is responsible for tens of thousands of human deaths annually, and in resource-constrained settings, vaccinating dogs to control the disease at source remains challenging. Currently, rabies elimination efforts rely on mass dog vaccination by the parenteral route. To increase the herd immunity, free-roaming and stray dogs need to be specifically addressed in the vaccination campaigns, with oral rabies vaccination (ORV) of dogs being a possible solution. Using a third-generation vaccine and a standardized egg-flavoured bait, bait uptake and vaccination was assessed under field conditions in Namibia. During this trial, both veterinary staff as well as dog owners expressed their appreciation to this approach of vaccination. Of 1,115 dogs offered a bait, 90% (n = 1,006, 95%CI:91–94) consumed the bait and 72.9% (n = 813, 95%CI:70.2–75.4) of dogs were assessed as being vaccinated by direct observation, while for 11.7% (n = 130, 95%CI:9.9–17.7) the status was recorded as "unkown" and 15.4% (n = 172, 95%CI: 13.4–17.7) were considered as being not vaccinated. Smaller dogs and dogs offered a bait with multiple other dogs had significantly higher vaccination rates, while other factors, e.g. sex, confinement status and time had no influence. The favorable results of this first large-scale field trial further support the strategic integration of ORV into dog rabies control programmes. Given the acceptance of the egg-flavored bait under various settings worldwide, ORV of dogs could become a game-changer in countries, where control strategies using parenteral vaccination alone failed to reach sufficient vaccination coverage in the dog population.

**Funding:** This research was funded by the German Ministry of Health under the Global Health Protection Program (GHPP, https://ghpp.de/de/projekte/onehealth-namibia/ - grant number ZMVI1 - 2520GHP701) to KD. The funders had no role in study design, data collection and analysis, decision to publish, or preparation of the manuscript.

**Competing interests:** I have read the journal's policy and the authors of this manuscript have the following competing interests: AV and SO are employees of the Ceva Innovation Center GmbH, Germany, a company manufacturing oral rabies vaccine baits for wildlife and dogs. GG is employee of SWAVET, Windhoek, Namibia, a commercial supplier of veterinary medicinals and instruments. All remaining authors declare no conflict of interest. The collection, analyses, and interpretation of data, the drafting of the manuscript, and the subsequent decision to publish was jointly made by all co-authors.

## Author summary

Rabies in dogs can be prevented by vaccination, and this approach has become a cornerstone in the control and eventual elimination of the disease. However, vaccinating hard-to-reach often free-roaming dogs are a challenge and represents one of the challenges to reach sufficient herd-immunity. A potential solution would be to vaccinate these dogs using oral baits filled with a vaccine. In this study we have assessed the acceptability of oral rabies vaccination (ORV) in Namibian dogs under field conditions. The results demonstrate that the method is acceptable both for the owners and the dogs, with a very high uptake of the egg-flavored bait. This supports the potential of ORV to contribute to vaccination programs where parenteral vaccination alone failed to reach sufficient vaccination coverage in the dog population.

## 1. Introduction

The Tripartite (WHO, OIE and FAO) considers rabies control a priority but also an entry point to strengthen the underlying systems for coordinated, collaborative, multidisciplinary and cross-sectoral approaches to the control of health risks at the human-animal interface [1]. Among the various mesocarnivorous and chiropteran rabies reservoir hosts [2,3], domestic dogs pose by far the greatest threat to global public health [4,5]. Mass dog vaccinations and public awareness are key to success. Vaccinating at least 70% of the targeted dog population would break the cycle of transmission within the dog population and from dogs to humans saving the lives of several tens of thousands of people [6]. While concerted control measures at national and supranational levels have been successful at eliminating dog-mediated rabies in upper-income countries in Europe and North America [7,8], over the past three decades Latin America and the Caribbean have made impressive progress in controlling the disease at the animal source [9,10]. In 2019, Mexico was the first country to declare freedom from dog-mediated rabies [11], while the remaining countries in this region are on the cusp of eliminating rabies deaths or even in the endgame of dog rabies elimination [12]. Despite these successes, dog-mediated rabies continues unabated in Africa and Asia and is responsible for an estimated 59,000 human deaths annually (95% CI 25,000–159,000) [13]. At present, parenteral vaccination is considered the only approach for addressing dog-mediated rabies at-scale, however, implementing these techniques in resource-poor settings can be challenging. There are increasing reports of the inadequacies of this approach among important subpopulations of susceptible dogs. Perhaps the greatest challenge is maintaining adequate herd immunity in free-roaming dog populations [14–16]. A promising alternative solution to this problem maybe oral rabies vaccination (ORV) [16,17].

For example, ORV has been successfully used in eliminating rabies in wildlife populations. Over the past 4 decades, due to large-scale ORV programs fox-mediated rabies has virtually disappeared in large regions of western and central Europe and Canada [18–20]. Using the same approach rabies epizootics in coyotes and gray foxes in the US could be brought under control [21]. While ORV has been a cornerstone in rabies virus elimination from wildlife populations, oral vaccines have never been effectively used in dog rabies control programs and are still an undervalued tool for achieving dog rabies elimination [16,17]. Although the WHO issued recommendations on ORV of dogs [22], the number of studies is still limited. A few oral rabies vaccine strains have been investigated for ORV in dogs under experimental or confined conditions [23–29].

Attractiveness and uptake of different baits developed for dogs have been tested before [30–39]. While immunogenicity studies in local dogs using different vaccine bait combinations have been conducted in among others Tunisia [40,41], Turkey [42], India [43], Namibia [44] and Thailand [45], at least one efficacy study met international standards applicable at that time [43]. However, only few field applications have been documented so far [42,46–50].

With the launching of the Global Strategic Plan for elimination of dog-mediated human rabies deaths by 2030 [51], the concept of ORV in dogs gained momentum again to be employed as a complementary approach to current, traditional mass dog vaccination efforts [52].This strategy is currently promoted by the WHO and the OIE [16], but with the exception of Thailand [50], field data on its applicability and effectiveness under various socio-economic settings are lacking. Presently, a dog rabies elimination program using mass vaccination campaigns is implemented in the Northern Communal Areas (NCAs) of Namibia where the percentage of owned but free-roaming dogs is high [53]. Also, follow-up investigations indicated that the vaccination coverage reached was below the thresholds needed for rabies control and elimination [54]. Therefore, we set out to implement an ORV pilot field study using a 3rd generation oral rabies vaccine with a high safety profile according to international standards to demonstrate the applicability of this approach in Namibia, potentially serving as a blueprint for other regions in Africa, and beyond, where dog rabies is still endemic and the accessibility of the target population is a key constraint. The objectives of this study were to test the feasibility and benefits of ORV in dogs as a potential complementary tool within the rabies programme in Namibia by assessing bait uptake and vaccination rate in Namibian dogs and the acceptance of the method by veterinary authorities and local dog owners.

## 2. Materials and methods

### 2.1. Ethics statement

The implementation of the ORV field trial was an integral part of the official national canine rabies control program under leadership of the Namibian Directorate of Veterinary Services (DVS) in the Ministry of Agriculture, Water, Forestry and Land Reform (MAWLR) [53,55]. Approval to use the non-licensed vaccine in the frame of a disease control trial was granted by the Chief Veterinary Officer of the DVS at the MAWLR, Namibia. Data from an immunogenicity study showing non-inferiority of the immune response after oral vaccination to parenteral vaccines in local Namibian dogs [44], a human risk assessment for the specific live-attenuated vaccine virus [56] and the submission of a detailed study plan to DVS were basic prerequisites for decision taking. Importation of the oral rabies vaccine was authorized by the Namibian Medicine Regulatory Council (NMRC) under section 31(5) (c) of the Medicines and Related Substances Control Act 2003—registration number: 17.12.20/PW/2021/IMPORT-L/0009/ek. Under this permission, the vaccine baits were imported via SWAVET Namibia.

Additional approval from the Namibian ethics committee was not required since no personal data of dog owners were obtained. Approval of the field trial by DVS was given under the premise that the purpose of this pilot field trial had to be explained to dog owners and that the dog owner had previously given his/her formal verbal consent that his/her dog(s) could be offered a vaccine bait. To this end, dog owners were given a specific leaflet with ORV related information provided in both the official (English) as well as the local (Oshiwambo) language and issued a certificate of bait consumption that also contained an emergency contact phone number in case of any adverse events.

## 2.2. Study sites

The ORV field trial was conducted in the NCAs, in different rural and suburban communities within the Omusati and Oshana regions (Fig 1). While in Omusati, more than 90% of the population live in rural areas, ppredominately living from subsistence crop and livestock farming, in Oshana, the economic centre of the north, 45% live in urban areas, and only for 13% of the population farming is the main source of income. The average population density in Omusati and Oshana is 9.39 people/km$^2$ and 35.77 people/km$^2$, respectively [57].

The field trial areas were selected after consultation with the Directorate of Veterinary Services (DVS) considering available infrastructure and logistics (Oshana—headquarters) and based on results of a Knowledge, Attitude and Practice (KAP) study conducted, indicating low vaccination coverage in certain regions due to free-roaming hard-to-reach shepherd dogs. These dogs accompany the movement of cattle herds, partly even across the border to Angola, and are often difficult to handle by their owners and vaccination teams.

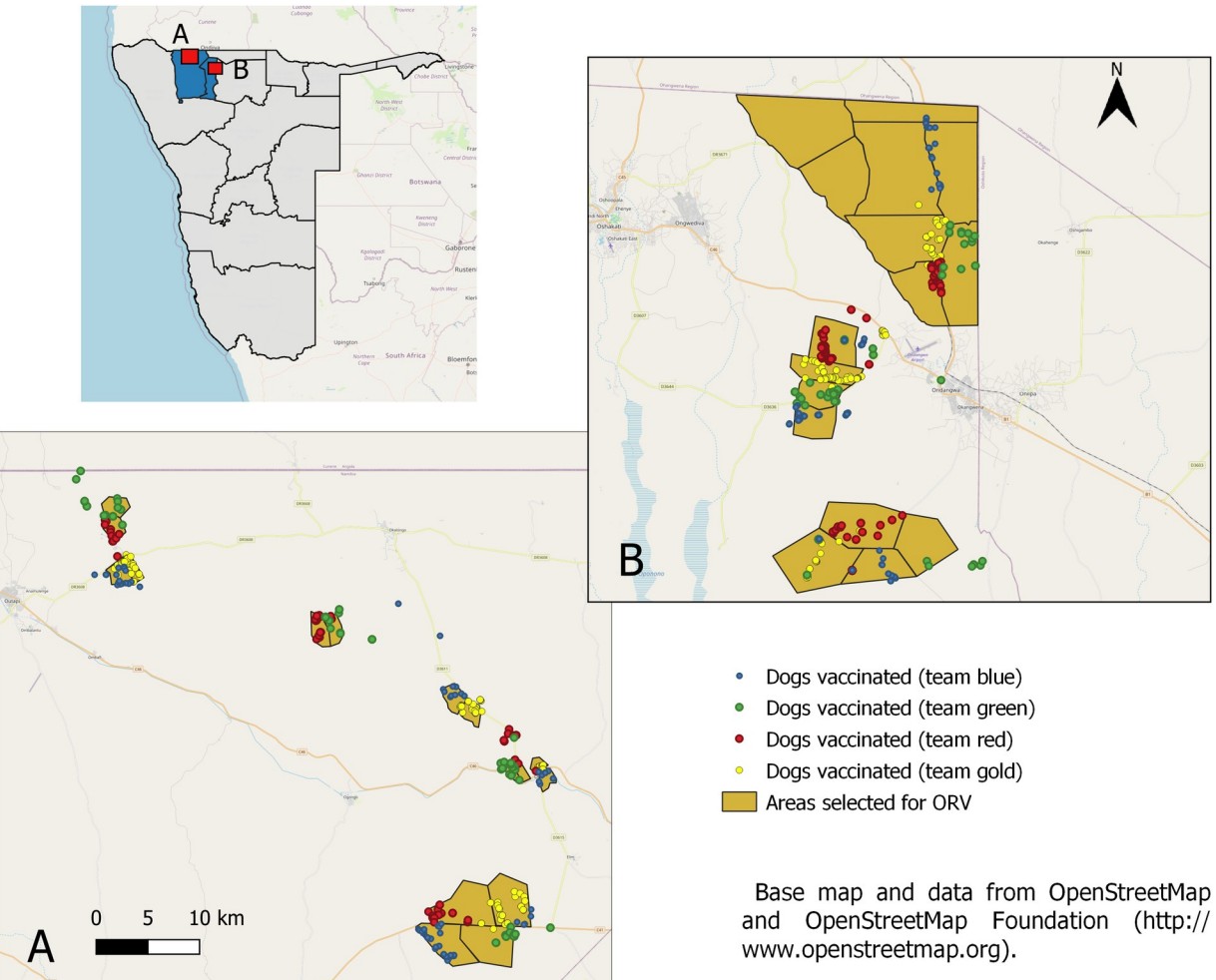

Dogs vaccinated (team blue)
Dogs vaccinated (team green)
Dogs vaccinated (team red)
Dogs vaccinated (team gold)
Areas selected for ORV

Base map and data from OpenStreetMap and OpenStreetMap Foundation (http://www.openstreetmap.org).

**Fig 1.** Map of Namibia (left) and the area of the field trial enlarged (right) for Omusati (A) and Oshana (B), with color-codes used for the individual teams. This map contains information from OpenStreetMap and OpenStreetMap Foundation (https://www.openstreetmap.org/#map=6/-23.544/17.842) which is made available under the Open Database License (https://www.openstreetmap.org/copyright).

## 2.3. Vaccine baits

Oral rabies vaccinations were conducted using 3rd generation oral rabies virus vaccine (Ceva Innovation Center GmbH, Dessau in Germany) consisting the SPBN GASGAS vaccine virus strain, a genetically engineered derivate of SAD L16 derived from the vaccine strain SAD B19 which is licensed for foxes and raccoon dogs according to international standards (Freuling et al., 2019). The recombinant vaccine virus construct is distinguished from SAD B19 by the deletion of pseudogene $\psi$, the introduction of four recognition sequences for restriction enzymes and duplicate insertion of an identical altered glycoprotein [58]. The genes encoding for glycoprotein G contain the amino acid exchange Arg333→Glu333 and Asn194→Ser194 to eliminate residual pathogenicity and reduce the risks for compensatory mutations, respectively [59]. These alterations, significantly enhance the safety profile of the vaccine virus [60]. A soft sachet filled with the liquid vaccine virus (3 mL, $10^{8.2}$ FFU/mL) was incorporated in a universal industrial manufactured egg-flavored bait (egg bait) previously shown to be highly attractive to local free-roaming dogs [38,39,61,62]. Immunogenicity of the vaccine baits had been demonstrated in local Haitian, Thai and Namibian dogs before [44,45,48].

Based on field experience, acceptance of the egg bait was further optimized by dipping them into locally available commercial tuna- or chicken liver-flavored cat liquid snacks immediately before offering the bait to the dog [50].

## 2.4. Shipment, transportation and storage of vaccine baits

Vaccine baits were shipped according to IATA guidelines on dry ice (UN 1845) directly from the manufacturer to the Central Veterinary Laboratory (CVL), Windhoek, using a commercial courier service. After temporary storage at CVL the vaccine baits were further transported to the Ondangwa branch of CVL located in the Oshana region of the NCAs. Upon arrival in Windhoek and at the field study areas the vaccine baits were stored in standard style freezers at -18 to -20˚C until further transportation or use in the field, respectively. Maintenance of the cold chain was ensured and documented using temperature data logger and integrated electronic measuring. Prior to shipment and the prior to start of the field trial, the quality of the baits and the vaccine titre was checked independently by the national and OIE reference laboratory at the Friedrich-Loeffler-Institute (FLI) essentially as described [63].

## 2.5. Vaccination teams

Immediately prior to the field trial, a two day staff introduction session and workshop was conducted during which staff was trained on the objectives of the field trial, oral rabies vaccination, vaccine bait handling, safety issues, techniques for approaching free-roaming dogs, best practice on offering vaccine baits to dogs, electronic data collection (bait handling by individual dogs—duration, consumption, perforation and/or swallowing of sachet), and interpreting effectiveness of vaccination attempt. The importance of retrieving the discarded vaccine sachet after bait consumption as described [50] was highlighted followed by a door-to-door vaccination training in the field.

There were four vaccination teams working simultaneously, with each team consisting of two DVS staff members (state veterinary officer, animal health technician), a data collector and an international expert. While the state veterinary officers were responsible for contacting dog owners, explaining the purpose of the study, seeking owners consent and issuing a certificate of bait consumption, the animal health technicians acted as vaccinators. Data collectors comprised faculty members and students from the Faculty of Agriculture and Natural Resources, Ogongo Campus, University of Namibia (UNAM). Vaccination teams used four-

wheel drive pick-up trucks equipped with coolboxes, cooling bags, gloves, rubbish bags, and disinfectants.

## 2.6. Vaccinations

Vaccination campaigns were announced via local radio the evening before and the morning the campaigns took place. Both door-to-door as well as central-point vaccinations were conducted. Vaccine baits were distributed to the targeted dog population using the hand-out and retrieve model [61]. The field trial was carried out at the end of the dry season during the second half of October 2021. During this time, vaccinations were performed over eight full working days (including two half days).

Vaccine baits were transferred to portable cool boxes the evening before field use, allowing them to thaw before they were offered to the dogs. Baits unused at the end of the vaccination day were kept at refrigerator temperatures (4–8˚C) and offered to dogs the next day to avoid repeated freezing and thawing of vaccine baits. Baiting was conducted both at individual homesteads as well as at central places in villages where people brought their dogs for oral rabies vaccination. Vaccination took place between 8:00 am and 6:00 pm. Team debriefings and daily evaluations were held at the end of each vaccination day.

Vaccination team members handing the baits (vaccinators) wore examination gloves. Dog owners were informed that dogs offered a bait should be left alone for 12 hours to minimize potential contact with the live vaccine virus. Any discarded sachet was retrieved, collected in trash bags and disposed of as infective materials at the Ondangwa branch of the CVL according to prevailing regulations on hazardous waste.

## 2.7. Data collection and vaccination monitoring

For collection of vaccination and survey data as well as project management, e.g. navigation within demarcated boundaries, sharing real-time team locations during roaming work and survey assessment, a smartphone application including the web-based backend platform was used essentially as described [64]. The App was provided by Mission Rabies, a non-governmental organization specializing in large scale rabies control (https://missionrabies.com/). Smartphones with WVS version of the Mission Rabies App installed were provided to each team. Survey related data including dates, owner consent, size (small = <10kg; medium = 10-30kg; large = >30kg), sex and number of dogs per household, dogs vaccinated and bait handling by individual dogs, i.e. duration, consumption, perforation and/or swallowing of sachet, and the resulting assumed vaccination status (vaccinated, non-vaccinated, unknown) were recorded on the phones using questionnaire forms, pre-designed by an administrator on the backend platform and remotely loaded to the handsets using 3G. Data were entered offline and stored locally on the handset where it could be reviewed on a map the same day. The app was also used to assign working zones for each vaccination team (different colours–gold, red, green and blue) on the App backend platform the day before with demarcated boundaries for each zone automatically synchronized to the App on each teams' handset via internet connection.

## 2.8. Evaluations and statistical analysis

A dog was considered 'interested' if the animal had any direct contact (smelling, licking) with the bait offered, irrespective of subsequent handling. Animals were regarded as successfully 'vaccinated' if the bait chewing and intensity (thoroughness) was detectable and/or perforation of the sachet clearly visible. Any dog that swallowed the bait immediately, or walked away with it and could not be observed, or chewed inappropriately on the bait without visible perforation

of the sachet was assigned an 'unknown' vaccination status. The status 'non-vaccinated' was assigned if the dog was not interested or the bait was only shortly taken up and immediately dropped with the bait casing and sachet still intact. The latter also applied to dogs that showed interest (and accepted the bait) but were interrupted by external factors (other dogs, humans, cars, etc) and discontinued bait handling. The maximum observation time for a dog was three minutes.

Data were uploaded daily to a cloud-based server and downloaded by evaluation supervisors as an Excel document Microsoft Excel 2013 (Microsoft Corporation, Redmond, WA, USA) for initial review and analysis. Spatial information was analyzed and displayed using QGIS Geographic Information System (QGIS.org, 2022.http://www.qgis.org) with base map and data from OpenStreetMap and OpenStreetMap Foundation (http://www.openstreetmap.org).

Statistical analysis was performed first by univariate contingency table testing (Chi$^2$—and Fisher's exact test) and followed by a multiple logistic regression (MLR). The dependent variable was "vaccination success" (yes/no), and datasets for dogs with an "unknown" status had to be removed. Independent variables were date, period of the day, team, level of supervision, if the dog was alone or together with other dogs, size and sex of dogs. Variables with $p \leq 0.20$ (univariate analysis) were included into the final MLR model. This cut-off value ($p = 0.20$) instead of the standard 0.05 was selected for the univariate analysis as recommended [65], the latter can fail in identifying variables known to be important in a multivariable analysis. Statistical analyses were carried out using GraphPad Prism v9.0 (GraphPad Prism Software Inc., San Diego, USA).

## 3. Results

An exceptionally high percentage of dog owners (99%) agreed to have their dogs vaccinated with this novel technique and vaccine bait. Of ten households contacted where vaccination was not conducted because of missing consent, in seven cases the owner was absent, in one case there was no person above 18 years of age available and two dog owners refused to get their dog vaccinated. Using the mobile planning and data capturing technology, a total of 1,139 datasets were generated.

The majority of dogs (78%) encountered and offered a bait during the study were owned and free-roaming. The proportion of ownerless free-roaming dogs was 3%, while the remaining dogs were assessed as confined during the vaccination. With 63%, there was a gender bias towards male dogs. Larger dogs (>30 kg) were rare (8%), whereas medium (57%) and small (<10kg; 33%) were dominating in the dog population. The majority of dogs (80%) were offered baits at the individual homesteads while the others were baited at central places (crush-pens, village centres, etc.) in respective areas (Fig 1).

The mean distance between individual baitings per team was 533m, with the lowest mean distance (226m) at the last day of the study when semi-urban areas were included. The longest distance between two baitings was 10km (Fig 2A). Overall, under field study conditions, the average number of dogs vaccinated per hour was 7, with a maximum of 28 dogs vaccinated per hour for one team (Fig 2B).

Of 1,115 dogs offered a bait, 93.6% (n = 1,044, 95%CI:92.0–94.9) were interested and 90% (n = 1,006, 95%CI:91–94) consumed the bait (Fig 3). Overall, 72.9% (n = 813, 95%CI:70.2–75.4) of dogs were assessed as being vaccinated, for 11.7% (n = 130, 95%CI:9.9–17.7) the status was recorded as "unknown" and 15.4% (n = 172, 95%CI: 13.4–17.7) were considered as being not vaccinated. In 54.9% (n = 552) of dogs observed, the vaccine blister was swallowed, while 43.4% (n = 437) of dogs that consumed a vaccine bait discarded the blister. For the remaining

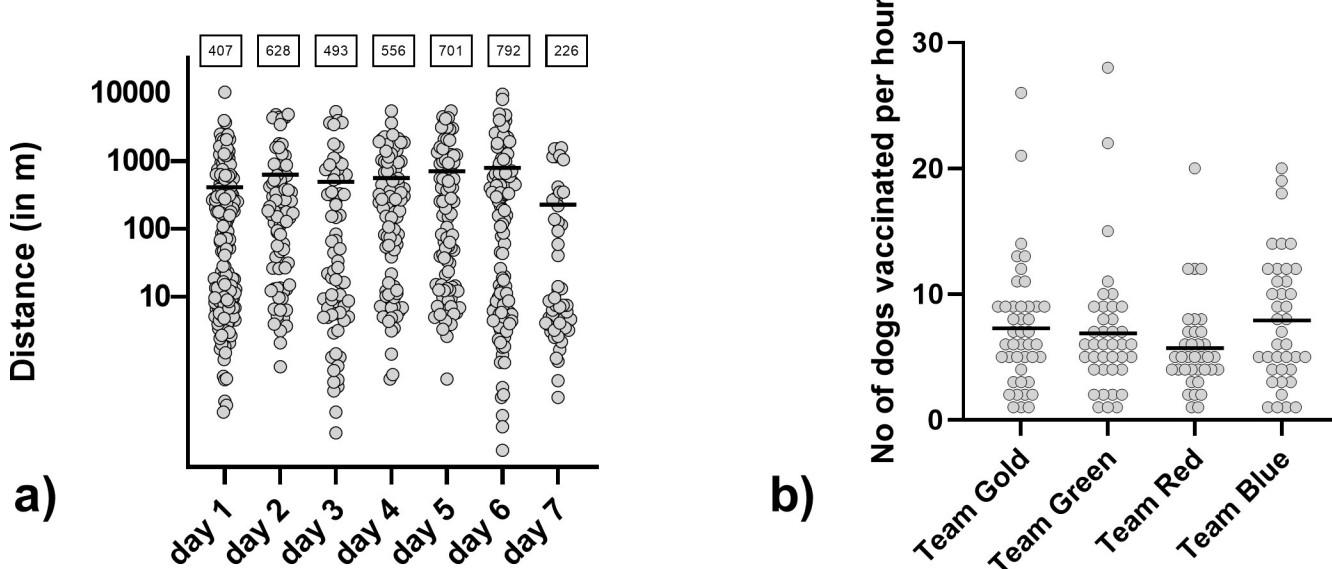

**Fig 2.** Euclidian distance between consecutively baited dogs per day as calculated by their individual GPS-tracked position (a), with the mean indicated (value shown in boxes). Number of dogs vaccinated per hour and team (b).

dogs, the status of the bait could not be verified, as e.g. the dog ran away with the bait and could not be observed anymore. Only 9.8% (N = 43) of all blisters retrieved were not perforated.

For the statistical analysis, 985 entries with a vaccination assessment (yes/no) were available. A statistically lower vaccination rate (p = 0.0048, Chi-square test) was observed on the last (69.8%) and first day (76.7%) of the campaign (Fig 4A). Differences in vaccination rates during time of the day (Fig 4B) and the different teams were not significant (Fig 4A).

While there was no statistical difference in vaccination status in regard to the confinement status (Fig 5B and S1 Table) or the sex of the dog (Fig 5B), smaller dogs (p = 0.0166, Chi-square test) and dogs offered a bait with multiple other dogs being present (p = 0.0494, Fisher's exact test) had significantly higher vaccination rates (Fig 5C and 5D). All variables with a p<0.20 identified, i.e. date, size and social situation of the dog, in the univariate analyses were included in a multivariate logistic regression model, but only size and social situation had a significant impact (S2 Table). Vaccination success was higher in small dogs and when more than one dog was present and was offered a bait.

The amount of bait matrix consumed did not affect vaccination success. However, the chewing time and fate of the sachet (discarded or swallowed) had a significant effect on vaccination success (Fig 6 and S3 Table). Dogs that chewed long (>60sec) and dogs that discarded the sachet were less likely to be considered vaccinated. Dogs chewing long rarely swallowed the sachet (12.4%), meanwhile most dogs that chewed for a very short time swallowed the sachet (74.2%).

## 4. Discussion

Overall, the results from this first ORV field trial in Namibia demonstrate a high acceptance for this method both by the veterinary/technical staff as well as the dog owners. In the field, the apparent efficiency in vaccinating dogs, particularly those that cannot be easily handled, was well acknowledged both by the veterinary staff involved as well as by the owners of dogs. For many dogs, this was the first time they had ever been vaccinated. Only very few individuals did

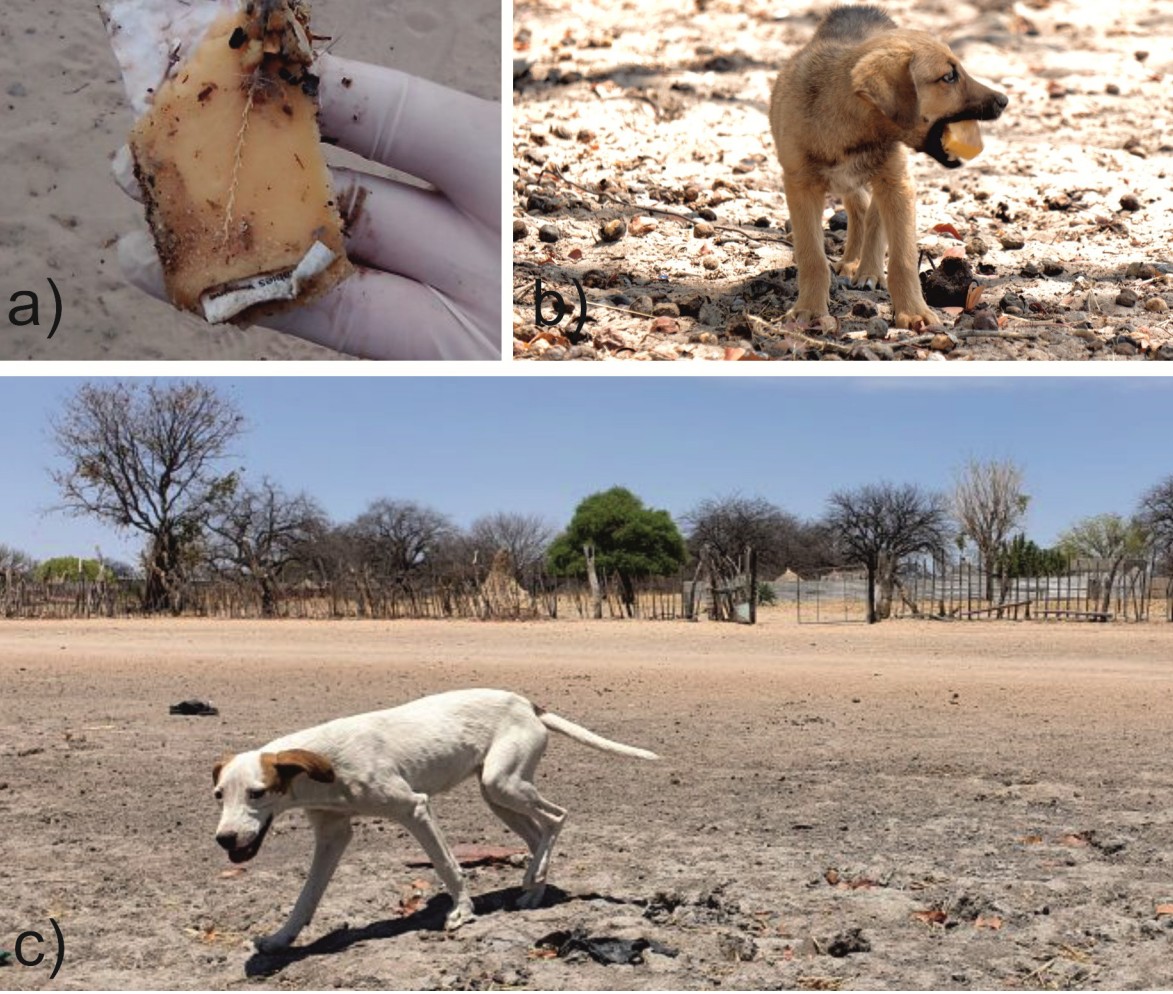

**Fig 3.** Remains of a partially consumed bait with the blister perforated (a). A puppy consuming a bait (b). Visual impression from a traditional homestead where dogs were vaccinated (c).

not give their consent to vaccinate their dog using a novel vaccination approach and a vaccine that is not yet licensed. This is surprising and very promising for future vaccination campaigns in dogs, as for human diseases there seems to be an increasing hesitancy for vaccination, e.g. for COVID-19 [66]. Public announcement prior to the campaign by radio, and the direct interaction with the dog owner by DVS likely played an important role in the acceptance of this approach.

The egg-flavoured baits, additionally dipped with meat-based flavor, were highly attractive to the dogs. This adds to the results of numerous studies, showing a high acceptance rate of ORV baits in dogs, e.g. Navajo Nations, US (77.4%) [38], Goa State, India (77.5%) [33], and Thailand (78.8%) [39] and Bangladesh (84%) [62]. The percentage of dogs offered a bait that were considered vaccinated by ORV in this field trial was at least 72.9% but likely higher, because a number of dogs disappeared with the bait and were considered "unknown". While about half of the vaccine blisters were swallowed, when blisters were retrieved, more than 90%

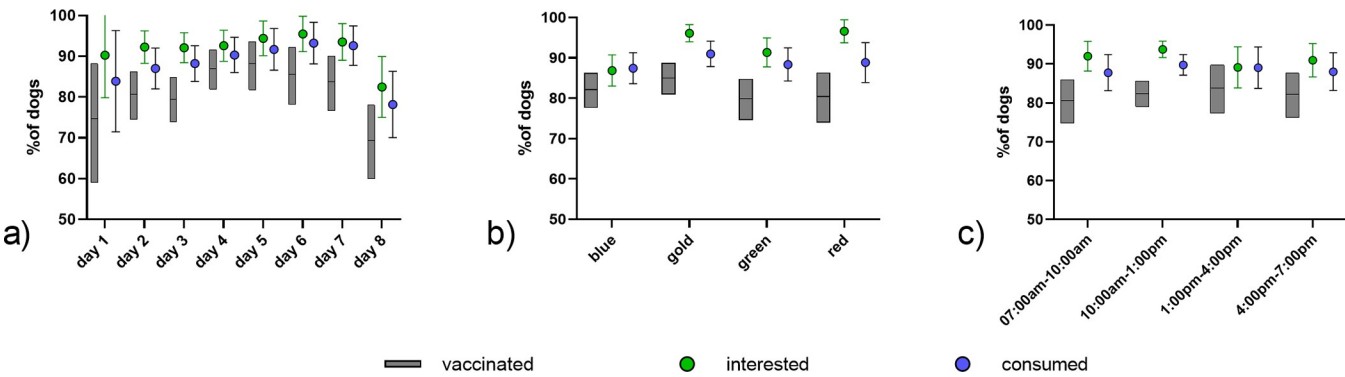

**Fig 4.** Comparison of bait interest, bait consumption and vaccination per study day (a), daytime (b), and team (c). The mean and the 95% confidence limits are indicated.

were perforated, suggesting that if the bait was consumed, a large proportion of dogs have likely had contact with the vaccine and can be regarded as vaccinated. In any case, the observed vaccination rate of dogs offered a bait was slightly lower than with the same bait in Thailand with 83% [39].

Although the assessment of vaccination was based on individual observation, the small differences between teams suggest that the overall bias was not affecting the outcome of the analysis (Fig 2A). Even though temperatures reached 35˚-39˚C during the early afternoon, this did not affect bait interest, consumption and vaccination success. ORV protocols for dogs in Namibia and likely in other areas with a similar situation can therefor disregard the time of campaign and focus on other parameters to increase effectiveness.

The fact that smaller dogs had a higher interest, consumed baits more readily and showed a higher vaccination rate as opposed to mid-sized and large dogs is interesting. Partly, these small dogs comprised of younger puppies (Fig 3).

Also, in situations when more dogs were around, smaller dogs tended to be more competitive towards consuming the bait, even though several baits were offered to avoid hierarchic feeding behavior. A similar observation was made in Thailand, where small and young dogs had higher bait acceptance rates [39]. Dogs that chewed long (>60sec) and dogs that discarded the sachet were more likely not vaccinated. Dogs chewing long rarely swallowed the sachet (12.4%), meanwhile most dogs that chewed very short swallowed the sachet (74.2%).

In the frame of this field trial with more than 1,100 baits handled, no adverse events were observed in dogs and vaccine exposures to humans that would require intervention did not occur. This adds to the high safety profile of this live vaccine when using the hand-out-and retrieve model [67]. Spillage of vaccine is not considered a source of contamination for potential contact to humans since the enveloped virus has a reduced viability in the environment. In the study area, the sandy floor, the high temperatures and the constant sunlight are further factors that decrease virus' persistence. We did not detect rabies cases in vaccinated areas in three months after vaccination.

There are some limitations to this study. For statistical reasons, datasets with vaccination status "unknown" had to be removed thus leading to higher proportions of dogs being interested, consuming the bait and being assessed as vaccinated than if they were included.

Because of the research character of this field trial, an assessment of the costs and efficiency of ORV as a tool under the Namibian settings cannot be made. For example, deep freezers for the storage of vaccines constituted more than one third of the entire budget for this project (Fig 7) but are one-time investments that may not be required in other settings, depending on

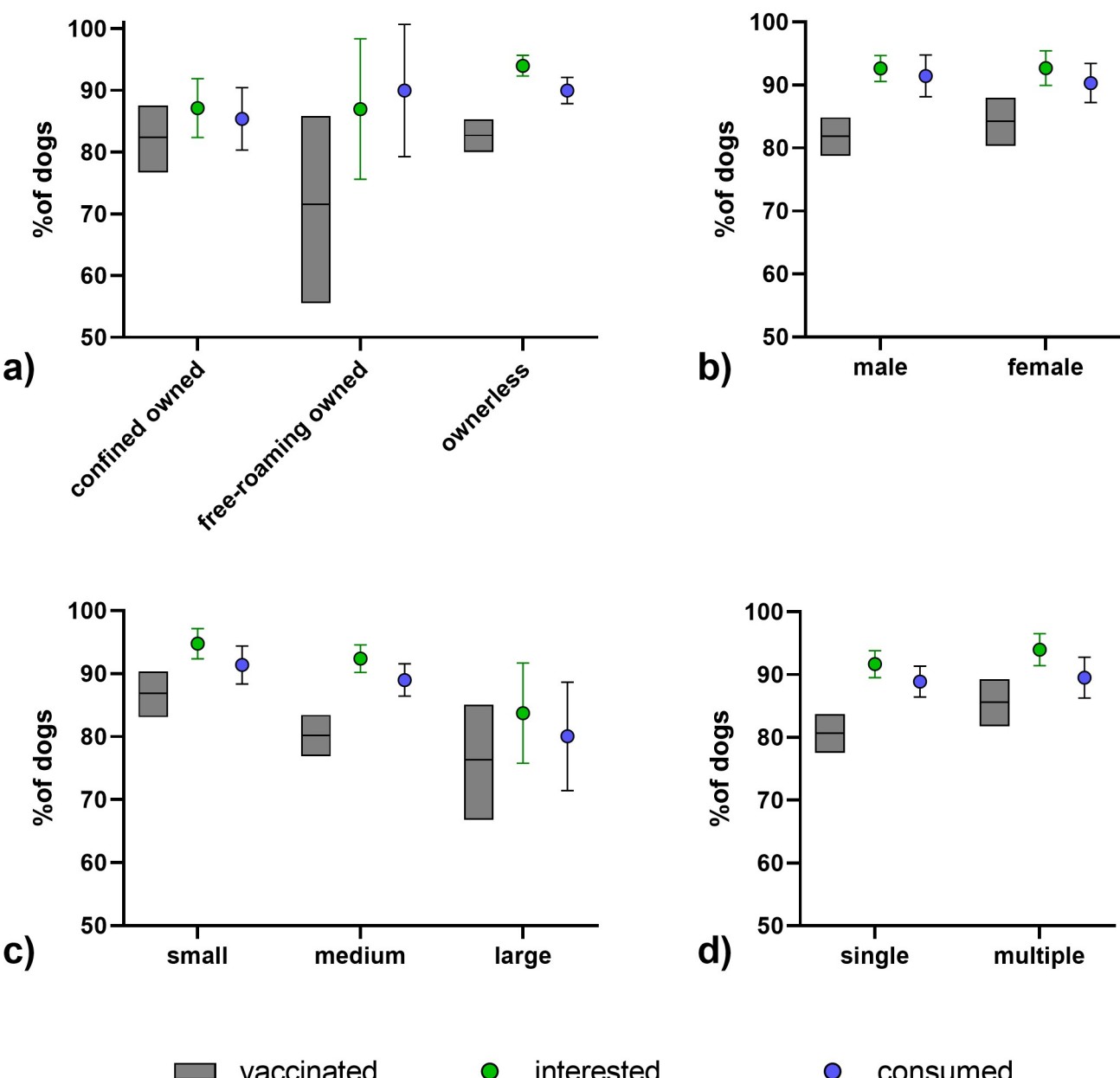

**Fig 5.** Comparison of bait interest, bait consumption and vaccination according to dog owner status (a), sex (b), size of the dog (c), and the social setting (d). The mean and the 95% confidence limits are indicated.

the prevailing logistical capacities and infrastructure. Also, due to the research component, more staff was involved than what would be needed if ORV was routinely used. In addition, accommodation and daily allowances were also provided to vaccinators; costs that may not be needed if regular staff is employed. As regards the costs for the oral vaccine baits, it is expected that prices could be reduced toward the minimum efficient scale if the market demand corresponds to the production capacities.

This research component with a required set of parameters to be typed into the mobile-phone app also prevented from vaccinating dogs in a shorter time interval when several dogs were presented for vaccination. Also, we did not attempt to vaccinate a certain proportion of

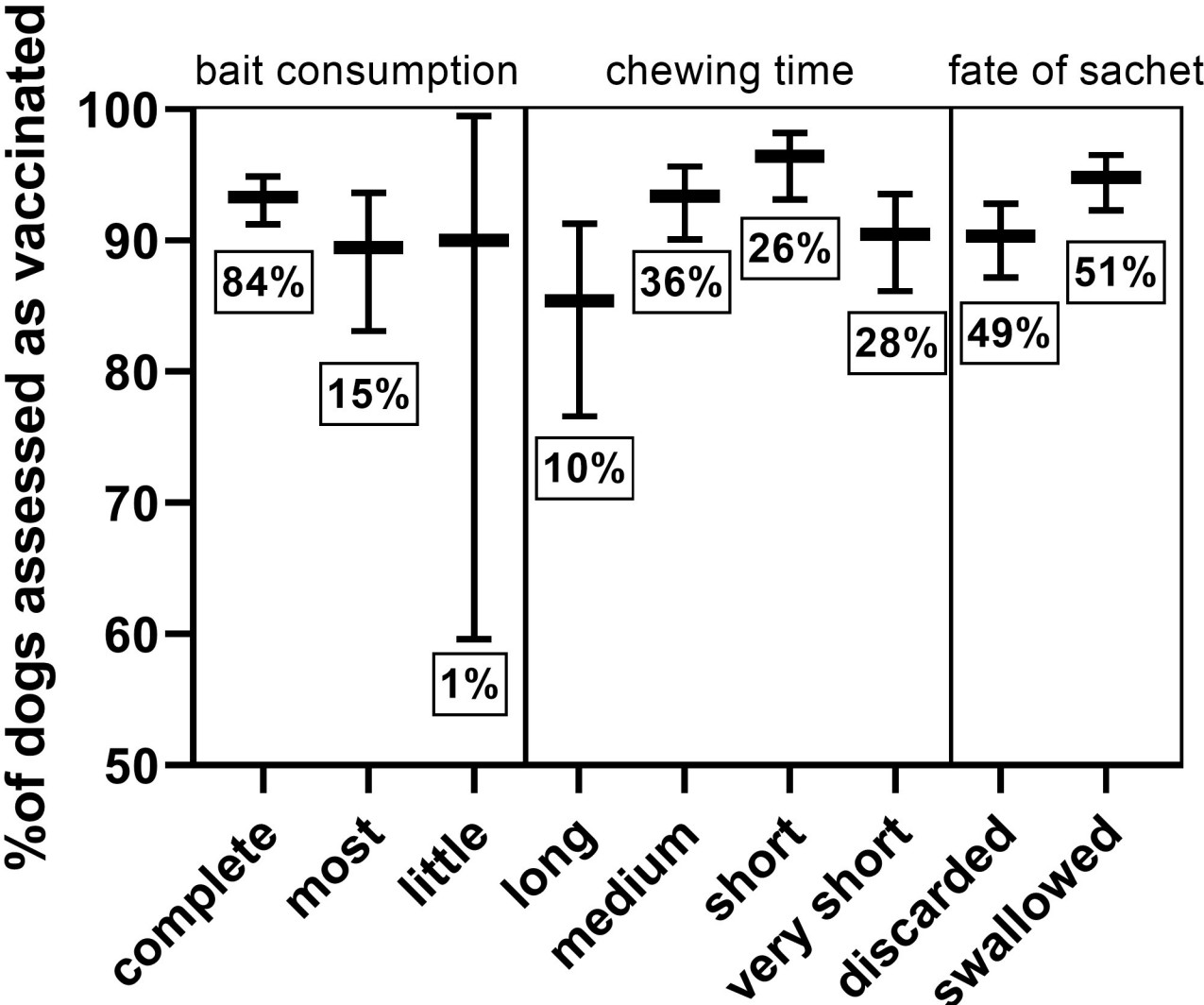

**Fig 6. Comparison of vaccination success according to bait consumption, chewing time, and the fate of the sachet.** The mean and the 95% confidence limits are indicated. The percentage of total dogs per assessment is given below each graph.

the dog population and all values given for vaccination rates refer to those dogs that were offered a bait, but not to the entire dog population.

One aspect that was identified to limit the potential of ORV in the field was the requirement of owners' consent prior to vaccination as was laid down in the study plan. Future campaigns should address this by indicating a general consent when the dog is free roaming at the time of vaccination. Another practical issue that emerged during the campaign was the provision of a vaccination certificate. Principally, the ORV method aims at the herd immunity and not the immune response in any individual dog, but specific ORV certificates may be issued during campaigns when ORV is included. In this field trial, both central-point vaccination as well as a door-to-door was used. As for the latter, with a highly dispersed human and dog population, partly absent dog owners, and distances between one and ten kilometers between individually vaccinated dogs (Fig 2C) if not even higher in other areas, this approach would be very inefficient and against the background of increasing costs for fuel, inappropriate under many settings. Rather, dog owners should be instructed to bring their dogs to a central point where

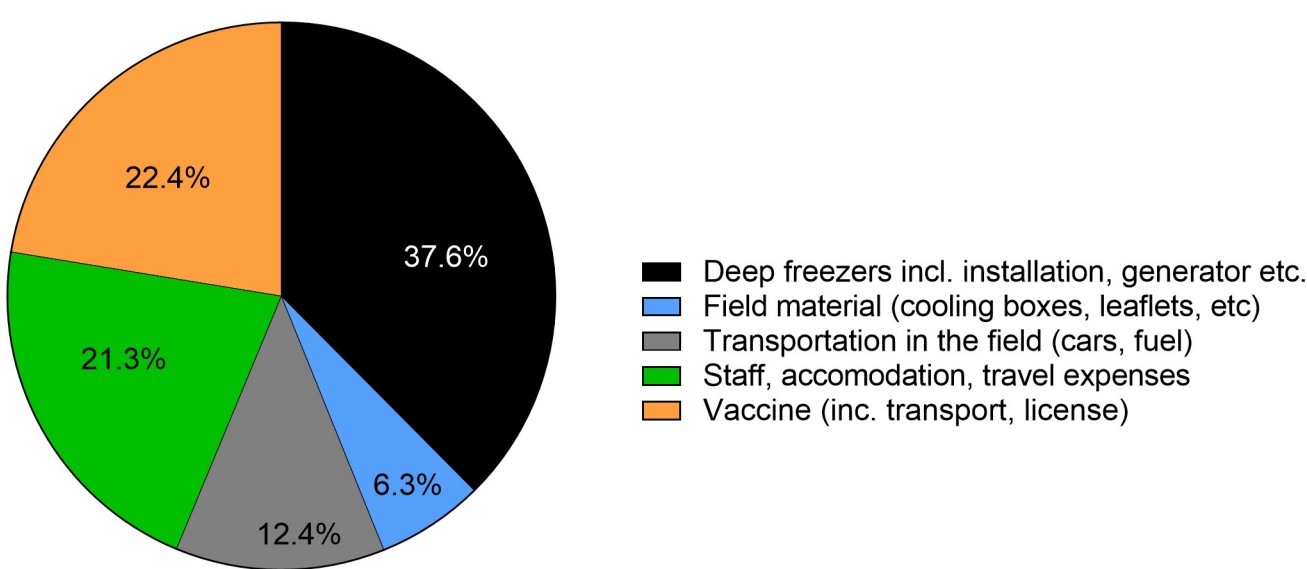

**Fig 7. Pie-chart showing the percent shares of different costs, with a total budget of 51.045 US$.**

parenteral and oral vaccination is conducted with a higher efficiency than parenteral alone. While dogs may be stressed due to the unfamiliar territory, other dogs and the transportation by leash as experienced before [44], in our study, we did not see a reduced bait uptake or vaccination rate when more dogs were present. However, to prevent negative influence dog owners could be instructed to keep their dogs at a certain distance.

In any case, a central point approach would again disregard those dogs that cannot be handled and brought to a vaccination point. To overcome this dilemma, the oral rabies baits could be handed over to the dog owners and vaccination would occur at their own premises, as has been demonstrated with non-vaccine baits in Tunisia [68]. A similar approach was also suggested for classical swine fever vaccinations to facilitate on-farm delivery in backyard pigs in remote areas [69]. For rabies, because of safety concerns such approach can only be envisaged for vaccines with a very high safety profile, so that a risk for humans is negligible [56]. While the vaccination success could not be controlled, this would still increase the herd immunity, particularly in the free-roaming hard-to-reach dogs. If dogs that act as super-spreaders are among those animals [70], targeting these highly connected dogs in the transmission networks would make vaccination campaigns more effective than random vaccination [71].

## 5. Conclusions

Even though planning and implementation of such a field trial in the midst of the COVID-19 pandemic represented a challenge, this pilot field trial of ORV in dogs in Namibia was very successful in terms of acceptance of the method, acceptability of the baits by dogs and the percentage of dogs offered a bait that were considered vaccinated. These results further support the strategic integration of ORV into dog rabies control programmes. Given the acceptance of the egg-flavored bait under various setting worldwide, ORV of dogs could become a game-changer in many African countries, where control strategies using parenteral vaccination alone failed to reach such vaccination coverage in the dog population that transmission was reduced and eventually controlled or eliminated, e.g. in West Africa [72], and Tanzania [15]. It is of note, that any study planning has to consider the availability of critical infrastructure to allow ORV baiting before the programme can be implemented.

Together with the recently published data on the epidemiology of rabies in Namibia [55], field data from dog vaccination campaigns [53,54], and immunogenicity of ORV in Namibian dogs [44] this study demonstrates Namibia's efforts in piloting and executing applied rabies research. Future research on best-practice examples should entail the parallel application of ORV (for inaccessible dogs) and parenteral vaccination at central vaccination points during i) mass dog vaccinations and ii) cattle vaccinations at crush pens. Additionally, the effectiveness of an optimized ORV-only approach with owner consent and limited data acquis needs to be assessed. Such research including cost-benefit analyses will provide evidence whether and how to integrate ORV into Namibia's rabies control programme.

## Supporting information

**S1 Table. The mean, minimum and maximum vaccination success rate (%) and the results of the univariate analysis of the selected independent variables (n = number of settings).** (PDF)

**S2 Table. Odds ratios of the Multiple Logistic Regression (MLR) Model.** (PDF)

**S3 Table. The effect of bait handling on vaccination success.** (PDF)

## Acknowledgments

The authors would like to thank all dog owners from the study areas in the Northern Communal Areas of Namibia for their overwhelming willingness to participate in this field trial. We also thank veterinary assistants Amanda Petrus, Taapopi Abisai, David Shaanika and Ashini Petrus for their participation in the vaccination teams and their excellent skills in vaccinating the local dogs. Special thanks go to students Hitjevi Mujorno, Ngurimuje Kahimuno, Abraham Kayambu and Sunny Nayeni from UNAM Ogongo Campus who served as data collectors in each of the vaccination teams. We are particularly indebted to Giulia Manzetti, who did an excellent job of documenting the entire field trial with photos and video recordings. Gratefully acknowledged is the GIS support by Patrick Wysocki and Ronald Schröder (FLI, IfE). Last but not least, the authors would like to thank Drs Gregorio Torres, Moetapele Letshwenyo and Tenzin Tenzin from the World Organisation for Animal Health (WHOA) for their continuous support of the Namibian-German collaboration and their encouragement and help in our joint research projects.

## Author Contributions

**Conceptualization:** Conrad Martin Freuling, Frank Busch, Adriaan Vos, Steffen Ortmann, Albertina Shilongo, Lukas Kaholongo, Thomas Müller.

**Data curation:** Conrad Martin Freuling, Adriaan Vos.

**Formal analysis:** Conrad Martin Freuling, Frank Busch, Adriaan Vos, Frederic Lohr, Albertina Shilongo, Jolandie van der Westhuizen, Thomas Müller.

**Funding acquisition:** Conrad Martin Freuling, Klaas Dietze, Thomas Müller.

**Investigation:** Conrad Martin Freuling, Frank Busch, Adriaan Vos, Steffen Ortmann, Frederic Lohr, Nehemia Hedimbi, Josephat Peter, Herman Adimba Nelson, Kenneth Shoombe, Albertina Shilongo, Brighton Gorejena, Lukas Kaholongo, Siegfried Khaiseb, Jolandie van der Westhuizen, Thomas Müller.

**Methodology:** Adriaan Vos, Steffen Ortmann.

**Project administration:** Frank Busch, Klaas Dietze.

**Resources:** Frederic Lohr, Goi Geurtse.

**Software:** Frederic Lohr.

**Supervision:** Frank Busch, Adriaan Vos, Kenneth Shoombe, Albertina Shilongo, Thomas Müller.

**Validation:** Thomas Müller.

**Visualization:** Conrad Martin Freuling.

**Writing – original draft:** Conrad Martin Freuling, Adriaan Vos, Thomas Müller.

**Writing – review & editing:** Conrad Martin Freuling, Frank Busch, Adriaan Vos, Steffen Ortmann, Frederic Lohr, Nehemia Hedimbi, Josephat Peter, Herman Adimba Nelson, Kenneth Shoombe, Albertina Shilongo, Brighton Gorejena, Lukas Kaholongo, Siegfried Khaiseb, Jolandie van der Westhuizen, Klaas Dietze, Goi Geurtse, Thomas Müller.

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
