## [Decision Letter · Decision Letter 0]

17 May 2022

Dear Dr. Freuling,

Thank you very much for submitting your manuscript "Oral rabies vaccination of dogs – experiences from a field trial in Namibia" for consideration at PLOS Neglected Tropical Diseases. As with all papers reviewed by the journal, your manuscript was reviewed by members of the editorial board and by several independent reviewers. The reviewers appreciated the attention to an important topic. Based on the reviews, we are likely to accept this manuscript for publication, providing that you modify the manuscript according to the review recommendations. 

Sincerely,

Ashley C Banyard, PhD

Deputy Editor

Ashley Banyard

Deputy Editor

Reviewer's Responses to Questions

**Key Review Criteria Required for Acceptance?**

**Methods**

-Are the objectives of the study clearly articulated with a clear testable hypothesis stated?

-Is the study design appropriate to address the stated objectives?

-Is the population clearly described and appropriate for the hypothesis being tested?

-Is the sample size sufficient to ensure adequate power to address the hypothesis being tested?

-Were correct statistical analysis used to support conclusions?

-Are there concerns about ethical or regulatory requirements being met?

Reviewer #1: Line 122: How were rural and suburban defined?

Lines 122 – 125: Please provide more information about these communities. What is human population? Estimated dog population? Area? Any significant cultural considerations?

Figure 1: Please confirm if these maps are open-access. Some google map baselayers cannot be published without written approval. 

Figure 1: The colors do not seem to provide any information relevant to the study results. If this is true, suggest to select just one color. If not true, please explain in a footnote and methods section. 

Figure 1: The sites are very difficult to view given how small and dispersed they are. Suggest to zoom in either by cropping the maps differently or having multiple inset images of these communities. 

Lines 151 – 153: was there evidence that egg-alone resulted in sub-par uptake? This is unfortunate, as it is one more logistical constraint for countries considering replicating this approach. 

Line 164: please add additional information about what assay was performed to validate the titer of the vaccines. 

Line 175: 4b4? Define

Section 2.4: what training did the vaccinators receive? What preventive measures did they receive in regards to working with a modified live vaccine? Were vaccination team members required to be vaccinated against rabies? Some of this information is found in section 2.5, but it seems more appropriate in section 2.4

Line 180 – 181: I am assuming these DD and CP methods used parenteral vaccines? Please clarify

Line 195: what is the recommendations if a bait is still unused after the second day? Refrigerate and re-use, or dispose?

Line 196 – 197: this statement is inconsistent with the described methods. If owners brought dogs to a central location, why were they given oral vaccines instead of parenteral vaccines?

Line 200: Per WHO and OIE recommendations, were these owners and community members also educated on what to do if an exposure were to occur, either through their recently-vaccinated-dog or a found-bait in the community?

Line 248: this is a very high p-value for inclusion in a final MLR model. Is there a citation for this approach, or at least commentary on why such a high inclusion cutoff was used?

Reviewer #2: The methods are generally clearly articulated but I have a few specific comments (below). The study design is appropriate to address objectives around the acceptability of oral vaccination (both to dogs and owners) and its potential value in scaling up mass dog vaccination in rural communities. 

It would be good to include in the abstract a concise description of how bait update and vaccination were assessed (e.g. direct observation) to clarify that no post-vaccination serological data were included in this analysis (which is something some readers may be expecting to see). 

Line 155, section 2.3. Provide a reference for methods used for checking quality of baits and vaccine titre. 

Fig. 1: More information needs to be provided in the legend with larger labels on the map to indicate locations, including the different community names. Given that >1,000 dogs were vaccinated, it’s not clear how the circles explicitly relate to individual dogs – presumably there are multiple circles that overlap? Some further information in the legend would be helpful. 

Section 2.5 Vaccinations. It would be useful to provide a description as to what type of central point locations were selected. Were these locations where you might expect to find ‘ownerless’ dogs?

How soon after vaccination were the discarded sachets retrieved?

Section 2.7 Evaluations and statistical analysis. Include how long (approximately) the animal was observed if it did not take the bait immediately. For example, if it walked away, was the bait retrieved immediately, or was the dog/bait observed for a specified period of time to see if the dog came back to it. 

Include some description of how free-roaming and ownerless dogs were identified and classified. Both 'free-roaming' and 'ownerless' seem quite difficult classifications to make if the evaluation was made while administering baits door-to-door or at a central point.

**Results**

-Does the analysis presented match the analysis plan?

-Are the results clearly and completely presented?

-Are the figures (Tables, Images) of sufficient quality for clarity?

Reviewer #1: Line 279: Is this the correct use of “datasets”? The sentence is confusing. 

Figure 2, panel (a) – showing this data on the log scale does allow for the full range of data to be presented, but its difficult for the reader to tell the median distance for each day due to presentation of data on the log scale. Suggest to find a way to present the median values, which is of more importance than the total (presented in the box). What is the reason to explore distance between dogs by day? Was there an a prior assumption that there would be a significant relationship? I do not understand why this association was explored and how it relates to the act of vaccinating dogs by oral route. 

Line 287: it is unclear. Were these confined dogs also given ORV? If so, why, as it seems like they could have been vaccinated through the parenteral route. 

Line 290: this idea of calling crush pens a “central point” vaccination may not be relatable to a majority of readership. It would be very interesting to have more explanation of this modified approach to central point, preferably detailed in the methods section under vaccination approaches. Otherwise, it appears like dogs were easily accessible to parenteral, but given ORV, which does not comply with the stated objectives. 

Line 291: that’s a long distance! I wonder if baiting by motorbike would improve efficiency. 

Line 299: “unkNown”

Line 296 – 299: What was the estimated vaccination coverage among the inaccessible, free-roaming dog population?

Line 303: were unperforated blisters re-used? Were these 45 vaccines reused?

Line 306 and 307: what were the a prior assumptions as to why time of day and day of campaign would be related to ORV acceptance? These seem like unrelated and random associations. Its not clear why they were explored and, had they been significant, likely would have represented spurious results. I strongly suggest to provide an explanation for some of these analyses in the methods, otherwise it appears like data fishing. 

Figure 3: Why does this figure use confidence intervals, while other figures with similarly presented data do not? The presentation of the data in the figures is inconsistent. That is not necessarily a problem, but seems unnecessary. 

Line 327: was “chewing very long” associated with size or age? It seems odd that chewing a long time would REDUCE the chance that a sachet was punctured. 

Line 328: “more likely not vaccinated” is confusing. Why not say, “less likely to be considered vaccinated”

Figure 5: Please clarify in the y-axis that this is the vaccination percent in the dog population. As presented now, it appears to be the percent of dogs, which cannot be more than 100% in each category. 

Results – what were the results of the sequencing from rabid dogs after the campaign (mentioned in methods? How many community vaccine contact events were reported? How many adverse events to vaccine were reported?

Reviewer #2: It would be good to include a breakdown of the number of dogs vaccinated door-to-door and those at central-point locations.

Line 327 and Fig 5: Consistency needed in terms of classification of the time baits were chewed e.g. the text refers to ‘very long’ (> 60 secs) but results are only provide for ‘long’ in Fig 5. Similarly, in the supplementary information 'long' refers to > 60 secs. 

I could not find reference to results in relation to rabies diagnosis and virus characterisation either in the main manuscript of supplementary tables. If no samples were obtained, I would suggest removing reference to this in the methods.

**Conclusions**

-Are the conclusions supported by the data presented?

-Are the limitations of analysis clearly described?

-Do the authors discuss how these data can be helpful to advance our understanding of the topic under study?

-Is public health relevance addressed?

Reviewer #1: Line 352: is this a fair comparison? Your egg baits were dipped in an additional meat-based sauce. Unfortunately, I believe this makes your comparison to the other studies inappropriate. It is, however, appropriate to note that ORV baits in numerous studies, including this one, have shown a high acceptance rate; particularly when they are tailored to local dog preferences. 

Line 355: this statement is misleading. You had a vaccination coverage of 73% among the dogs that were offered a bait. This is not the same as a vaccination coverage in the community. The study does not appear to have assessed community vaccination coverages achieved by this campaign. How many dogs are in the community? Is there a census? Was a post-vaccination survey conducted to determine the vaccination coverage?

Line 361: it is odd that body condition score was not collected during this study. Other papers have speculated that lower BCS may relate to better uptake due to hunger and reliance on scavenging for food resources. Why was BCS not collected? And could this be a possible explanation for the difference in uptake from Thailand?

Line 369: Why was uptake by peri-urban vs rural not explored in this analysis? The results indicate that it was part of the data collection process, but none of these results are presented. The discussion indicates that success was lower in peri-urban areas, but there is no data presented to share this. This would be a significant finding, much more so than vaccinations by day or team or hour. Strongly suggest that this is better explored in the analysis and discussed. 

Lines 371 – 380: the authors should clarify what is mean by small, medium and large. The methods indicate this was weight-based (kg). Could “small” dogs not just be thinner and therefore more aggressive to receive food? Minor clarification/explanation in the methods and here would be helpful to the reader. 

Lines 377 – 380: these sentences are just repeating the results and offer no insight into why this is relevant to the study or future oral vaccination campaigns. Personally, I don’t see the relevance at all, but if the authors think this is a noteworthy finding, it is not apparent as-written. I suggest to drop this analysis, or better explain why this is anything more than a spurious association. 

Line 390: the lack of considerations for costs and logistics make any claims about the feasibility of ORV in Namibia moot. Plenty of approaches can be enacted to improve vaccination coverages, but we do not apply them because they are too costly or logistically challenging. There is plenty of great data in this paper to suggest further exploration of the role of ORV in Namibia’s dog vaccination program. But this study should avoid any suggestion that the data shows that ORV in Namibia should be incorporated into the current strategy. Without cost-effectiveness considerations and consideration for the availability of critical infrastructure to allow ORV baiting, this study cannot claim the approach should be implemented. 

Line 431: this statement is misleading. The vaccination coverage of dogs in this community was not assessed. The authors have assessed the vaccine uptake among dogs presented to them. This should be clarified throughout. No effort appears to have been undertaken to estimate the community vaccination coverage after this campaign. 

Lines 432 – 436: statements such as these would be MUCH more impactful if a simple cost component were added to this paper. In the absence of considering cost, the interpretation of impact is severely limited.

Reviewer #2: The study was carried out as a campaign that only involved oral vaccination, which has generated some valuable data, but it does have important implications in terms of interpretation and generalisability. The discussion includes consideration of some of the limitations of the study design, but a key point that has not been fully discussed is that many of the dogs reached through the oral vaccination approach may have been accessible through parenteral vaccination, particularly after the community sensitisation and advertising activities that were conducted. So while the study clearly demonstrates acceptance of dogs and owners and the potential value of this type of approach, the discussion does not address explicitly whether or how the data provides insight on the coverage that might be expected to be achieved in ‘hard-to-reach’ rural dog populations or the ‘supplementary’ value of oral vaccination over and above parenteral vaccination.

**Editorial and Data Presentation Modifications?**

Reviewer #1: see comments above

Reviewer #2: A few very minor editorial modifications suggested:

Line 96. Shift ‘also’ to mid part of the sentence e.g. ‘Attractiveness and update ….have also been tested’. I would perhaps also add a phrase at the end of the sentence to indicate where/in what settings these have been tested. 

Line 99: ‘Studies’ should be ‘study’

Line 108: ‘where THE percentage’

Line 173: Remove ‘of’ following ‘comprised’

Line 329: Include ‘chewed FOR A very short time…’

**Summary and General Comments**

Reviewer #1: OVERALL: This paper describes the continued application of research findings to improve dog vaccination coverage in Namibia. The evolution of the Namibian rabies control program is exciting and provides an example for success in the region. The logistical efforts undertaken to conduct this study are much-appreciated and the results provide more evidence to the literature that ORV is a safe and effective method of vaccinating free-roaming dogs. While the paper is easy to read and is a very important topic, I have several concerns with the analysis and presentation of the data. These are detailed below, but summarized here as well:

1) Several components of the analysis seem to be rather meaningless and the authors do not adequately describe why they had any a priori interest in these assumptions. As expected, the results were not significant. If the associations have no plausible reason to be related, and the analysis shows they are not related… what is the point? This applies specifically to analyses related to time of day, day of campaign, distance by day, and chewing time. These topics, specifically, should be much better explained and rationalized if the authors feel they are important to remain in the study. 

2) Without consideration for cost, the authors are severely limited in statements suggesting that this is a feasible approach to dog vaccination. Even a simple table with costs incurred to operate this campaign would be incredibly useful for the rabies community and for understanding the logistical and cost barriers to expanding ORV use. Without a cost component, the authors will need to be very careful about over-interpreting the impact of these findings. 

3) The authors did not use egg baits. They used egg baits dipped in meat sauces. While interesting, this diminishes the comparability to other studies. Use caution when making such comparison statements in the article. 

4) the authors did not assess the post-campaign vaccination coverage in this community. Many statements could easily be taken out of context to imply that this campaign achieved >70% vaccination coverage. The authors methods and analysis only imply that 70% of dogs offered a bait were vaccinated. If the authors are certain that they approached EVERY dog in the community, then this claim may be appropriate. Unless this can be clearly and confidently stated, then the coverage reported here is not equivalent to the community vaccination coverage.

Reviewer #2: This is a well-written manuscript that presents important data on oral rabies vaccination of dogs, which is an area of growing interest in relation to scaling up of mass dog vaccination in order to reach international targets of zero human deaths from canine rabies by 2030 (“Zero by Thirty”. The study has been executed well and is suitable for publication. A few minor comments/suggestions are included in the sections above.

PLOS authors have the option to publish the peer review history of their article (what does this mean?). If published, this will include your full peer review and any attached files.

Reviewer #1: No

Reviewer #2: No

Figure Files:

Data Requirements:

Reproducibility:

References

---

## [Editor Report · Decision Letter 1]

24 Jun 2022

Dear Dr. Freuling,

We are pleased to inform you that your manuscript 'Oral rabies vaccination of dogs – experiences from a field trial in Namibia' has been provisionally accepted for publication in PLOS Neglected Tropical Diseases.

Best regards,

Ashley C Banyard, PhD

Deputy Editor

Ashley Banyard

Deputy Editor

The authors have addressed all queries raised and the manuscript is now suitable for publication.

---

## [Editor Report · Acceptance letter]

16 Aug 2022

Dear Dr. Freuling,

We are delighted to inform you that your manuscript, "Oral rabies vaccination of dogs – experiences from a field trial in Namibia," has been formally accepted for publication in PLOS Neglected Tropical Diseases.

Best regards,

Shaden Kamhawi

co-Editor-in-Chief

Paul Brindley

co-Editor-in-Chief
